# Ribosome recycling is coordinated by processive events in two asymmetric ATP sites of ABCE1

Elina Nürenberg-Goloub, Holger Heinemann, Milan Gerovac, Robert Tampé

Ribosome recycling orchestrated by ABCE1 is a fundamental process in protein translation and mRNA surveillance, connecting termination with initiation. Beyond the plenitude of well-studied translational GTPases, ABCE1 is the only essential factor energized by ATP, delivering the energy for ribosome splitting via two nucleotide-binding sites by a yet unknown mechanism. Here, we define how allosterically coupled ATP binding and hydrolysis events in ABCE1 empower ribosome recycling. ATP occlusion in the low-turnover control site II promotes formation of the pre-splitting complex and facilitates ATP engagement in the high-turnover site I, which in turn drives the structural reorganization required for ribosome splitting. ATP hydrolysis and ensuing release of ABCE1 from the small subunit terminate the post-splitting complex. Thus, ABCE1 runs through an allosterically coupled cycle of closure and opening at both sites, consistent with a processive clamp model. This study delineates the inner mechanics of ABCE1 and reveals why various ABCE1 mutants lead to defects in cell homeostasis, growth, and differentiation.

## Introduction

mRNA translation by the ribosome is a cyclic process, essential and conserved among all phyla of life (Ramakrishnan, 2002; Jackson et al, 2010; Dever & Green, 2012; Nürenberg & Tampé, 2013). The ribosome is composed of a small (30S/40S in Pro/Eukarya) and a large subunit (50S/60S), both of which recruit a variety of additional factors during the four steps of translation: initiation, elongation, termination, and recycling. The latter process implies splitting of ribosomal subunits after canonical termination (Pisarev et al, 2010; Barthelme et al, 2011; Shoemaker & Green, 2011) and is further linked to mRNA surveillance, ribosome-based quality control, ribosome biogenesis, and cell metabolism (Pisareva et al, 2011; Shoemaker & Green, 2011; Becker et al, 2012; Dever & Green, 2012; Strunk et al, 2012; Kashima et al, 2014; Preis et al, 2014; van den Elzen et al, 2014; Shao et al, 2015; Young et al, 2015). In Archaea and Eukarya, the key factor for ribosome recycling is the ATP-binding cassette (ABC)

protein ABCE1, also termed ribonuclease L inhibitor 1 (Rli1p) in yeast and PIXIE in *Drosophila melanogaster* (Coelho et al, 2005).

ABCE1 is a soluble twin-ATPase, which utilizes ATP to remodel large ribonucleoprotein complexes. This multi-domain molecular machine is one of the most conserved proteins in evolution and essential in all organisms investigated. The N-terminal domain harbors two diamagnetic [4Fe-4S]$^{2+}$ clusters (FeS) (Barthelme et al, 2007, 2011; Karcher et al, 2008). Two head-to-tail oriented nucleotide-binding domains (NBDs) align two nucleotide-binding sites and perform a tweezer-like motion upon ATP binding and hydrolysis. The mechanochemical energy is transferred from the NBDs to the associated FeS cluster domain, which swings out and splits the ribosome (Kiosze-Becker et al, 2016; Heuer et al, 2017). Many ABC proteins are asymmetric and possess one consensus and one degenerate site. The latter harbors mutations in conserved motifs essential for ATP hydrolysis. However, our knowledge of their mechanism is limited and multiple scenarios can be derived based on the structural and functional diversity of asymmetric ABC-type machines.

Ribosome recycling can be subdivided into sequential events: (i) binding of ABCE1 to a post-termination complex (post-TC) yielding a pre-splitting complex (pre-SC), (ii) ribosome splitting, (iii) formation of the post-splitting complex (post-SC) composed of 30S/40S·ABCE1, and (iv) ABCE1 release from the small ribosomal subunit. During the first step of ribosome recycling, ABCE1 binds the post-TC in the vicinity of the canonical GTPase control center and contacts release factor 1 (e/aRF1) or its homologs (Becker et al, 2012; Preis et al, 2014; Brown et al, 2015). The release factor in the A-site and ATP are indispensable for the subsequent ribosome splitting step (Pisarev et al, 2010; Barthelme et al, 2011; Shoemaker & Green, 2011). Thereafter, ABCE1 remains bound to the small subunit and may connect the post-SC to canonical translation initiation before it dissociates (Nürenberg & Tampé, 2013; Heuer et al, 2017; Schuller & Green, 2017). Despite important structural snapshots of pre- and post-SCs, characterized by extreme conformational changes, the molecular mechanism of ABCE1 remains enigmatic. Key questions are of special interest for our understanding of ribosome recycling: How do the two asymmetric nucleotide-binding sites in the ribosome recycling factor coordinate the process of ribosome binding, splitting, and release? Is ribosome splitting driven by ATP binding or hydrolysis?

---

Institute of Biochemistry, Biocenter, Goethe University Frankfurt, Frankfurt a.M., Germany

Correspondence: tampe@em.uni-frankfurt.de

 

Here, we delineate the mechanistic framework of ABCE1 in ribosome recycling. We identify a low ATP turnover, control site II and a high ATPase, power-stroke site I and define their distinct roles in ribosome binding and splitting. Successive domain re-organization in ABCE1 schedules the recognition of post-TCs, ribosome splitting, and formation of the post-SC disqualified for reassociation. Sequential occlusion of ATP in the two asymmetric sites powers conformational changes within the two NBDs and the movement of the FeS cluster domain. An allosteric crosstalk controls the nucleotide-binding sites, hence ABCE1 must harbor intrinsic checkpoints to regulate the progression of ribosome recycling.

## Results

### Two nucleotide-binding sites operate asymmetrically in ABCE1

To investigate the role of the two nucleotide-binding sites in ribosome recycling, we used *Sulfolobus solfataricus* ABCE1, which shares 42% and 44% sequence identity with the yeast and human ortholog, respectively. We substituted conserved residues required for ATP occlusion or hydrolysis to study the role of each nucleotide-binding site (Fig 1A and B). The conserved glutamates E238 (site I) and E485 (site II) adjacent to each Walker B motif act as a catalytic base for ATP hydrolysis. Therefore, single and double alanine substitutions (E238A, E485A, and E238A/E485A) lead to stable ATP occlusion in the corresponding sites by preventing ATP hydrolysis. Equivalent substitutions are known to deactivate ATP hydrolysis in other ABC proteins and arrest the fully closed conformation of their NBDs (Urbatsch et al, 2000; Smith et al, 2002; Hopfner & Tainer, 2003). In an opposite approach, we introduced a bulky, positively charged residue into the ABC-signature motif to prevent nucleotide occlusion in site I (S461R) or/and II (S214R). The structural and functional role of the ABC-signature motif for nucleotide occlusion has been elaborated previously (Smith et al, 2002; Szentpetery et al, 2004). All ABCE1 variants were purified to monodispersity with a characteristic absorption at 410 nm (Barthelme et al, 2007), demonstrating fully assembled FeS clusters (Fig S1).

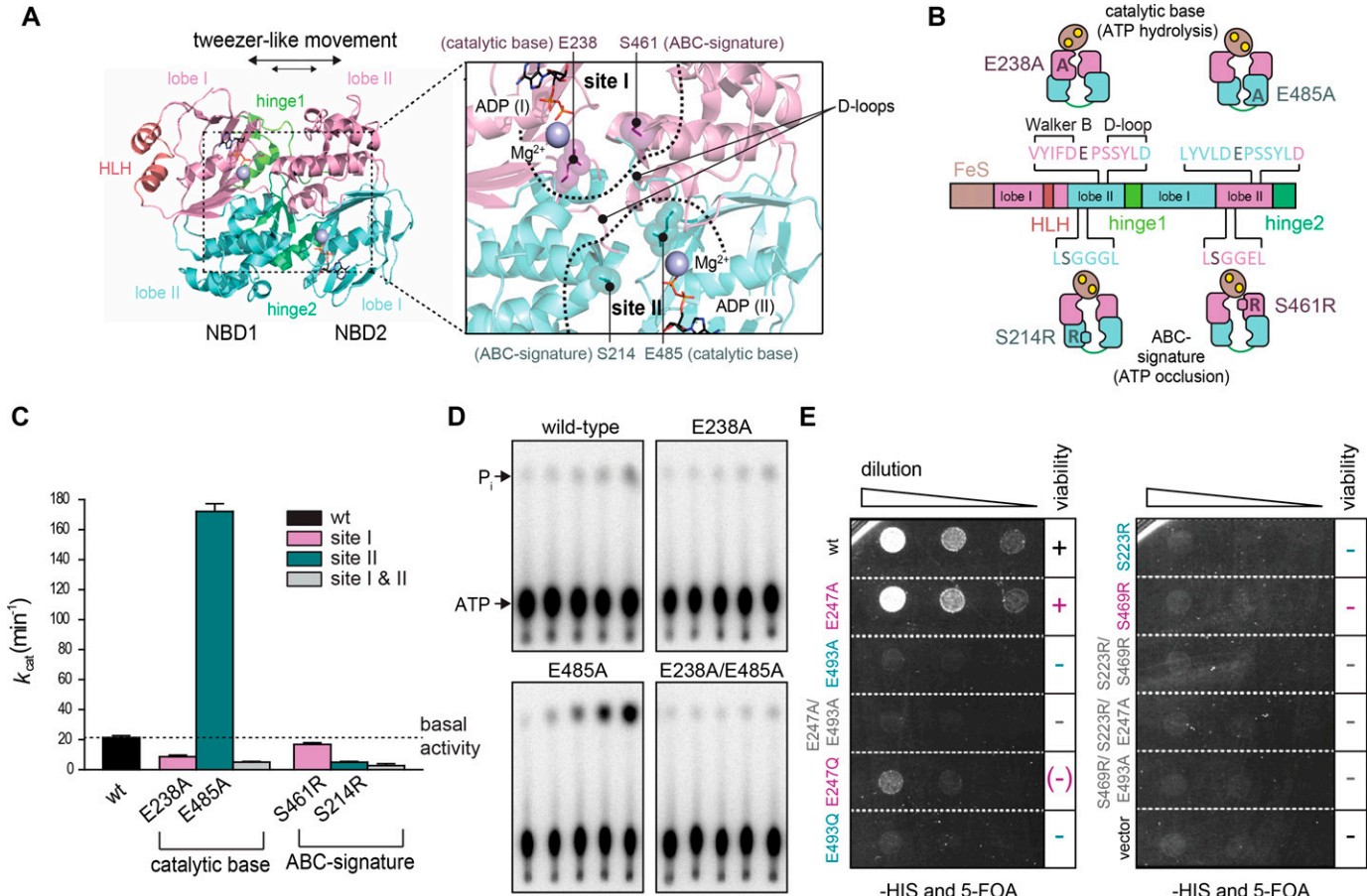

**Figure 1. Two nucleotide-binding sites of ABCE1 act functionally asymmetric.**
**(A)** Overall structure of ABCE1 without the FeS cluster domain (PDB 3OZX) and zoom into the two catalytic sites. Catalytic glutamates (E238, magenta, in site I; and E485, cyan, in site II) are located between Walker B and D-loop motifs. The ABC-signature motif approaches bound nucleotides from the opposing NBD and contains S214 (cyan, in site II) and S461 (magenta, in site I). **(B)** Strategic substitutions in ABCE1 prevent ATP hydrolysis or occlusion in the respective sites. **(C)** Asymmetric ATPase activity of ABCE1 mutants. Mean ± SD, n = 3. **(D)** Hydrolysis of $^{32}$P-γ-ATP (5 mM) by ABCE1 (1 μM) at 70°C. Representative set of three independent experiments. **(E)** Yeast plasmid shuffling assay illustrates the significance of an intact control site II. Only WT and ABCE1$^{E247A}$ (site I) remain viable (+), ABCE1$^{E247Q}$ (site I) shows a strong growth defect (–), whereas all other mutations are lethal (–). Representative set of two independent experiments.

The ABCE1 variants reveal a functional asymmetry with two distinct intrinsic ATP hydrolysis rates (Figs 1C and D, and S2), consistent with previous studies of related mutants (Barthelme et al, 2011). Inactivation of site I by mutations E238A or S461R leads to reduced ATP turnover carried out solely by a low-turnover site II. In contrast, ABCE1$^{E485A}$ is hyperactive, indicating that ATP occlusion in site II allosterically activates a high-turnover site I. Moreover, the disengagement mutation in site II of ABCE1$^{S214R}$ impairs the ATPase activity. Hence, ATP occlusion in site II is a prerequisite for ATP hydrolysis in site I. Owing to its allosteric impact on site I, the low-turnover site II is named control site. Notably, ABCE1$^{E485A}$ displays an eightfold increased activity (172 ATP/min) compared with WT ABCE1 (21 ATP/min), resembling the stimulated ATP hydrolysis of ABCE1 in the presence of splitting competent ribosomes and release factors (Pisarev et al, 2010; Shoemaker & Green, 2011). As negative control, ATP hydrolysis was strongly inhibited by substitution of the catalytic bases or disengagement mutations in both sites. The significance of an intact site II for ABCE1 function and viability of eukaryotic cells is emphasized by plasmid shuffling experiments in yeast (Fig 1E). Substitutions of the catalytic base (E493A and E493Q) or the disengagement mutation (S223R) in site II are lethal, whereas the site I mutants E247A or E247Q are viable. ABCE1 variants carrying mutations in both sites are lethal. A dominant negative growth effect was observed for the site II mutants E493A/Q and S223R, the site I disengagement mutant S469R, and the double-EA mutant (Fig S3). Hence, these mutants poison mRNA translation, for example, by constant occupation of the small ribosomal subunit, as shown in yeast (Dong et al, 2004) and *D. melanogaster* (Andersen & Leevers, 2007).

### Site II controls pre-SC formation

We next addressed the distinct roles of the two nucleotide-binding sites during formation of the pre-SC. In classical termination or mRNA surveillance, e/aRF1 or e/aPelota are delivered to the A-site by the GTPases eRF3, Hbs1, or aEF1α in Eukarya and Archaea, respectively (Carr-Schmid et al, 2002; Atkinson et al, 2008; Chen et al, 2010; Kobayashi et al, 2010; Brown et al, 2015). After GTP hydrolysis, eRF3, Hbs1, or aEF1α dissociate, leaving a splitting-competent post-TC. Owing to the fact that *S. solfataricus* 70S ribosomes are intrinsically instable and cannot be isolated by sucrose density gradient (SDG) centrifugation (Londei et al, 1986; Barthelme et al, 2011), we used ribosomes from *Thermococcus celer*. The high evolutional conservation of the translational machinery allows *S. solfataricus* ABCE1, aRF1, aPelota, and aIF6 to be functional with *T. celer* ribosomes (Barthelme et al, 2011). In the first set of experiments, we applied conditions non-permissive for 70S splitting to preserve the pre-SCs (25°C, 50 mM Mg$^{2+}$). ABCE1 requires aRF1 to efficiently form the pre-SC (Figs 2 and S4). WT ABCE1 specifically binds ribosomes in the presence of adenylyl-imidodiphosphate (AMP-PNP) but not ADP, confirmed by co-sedimentation with aRF1 (Fig 2A). ABCE1$^{E238A}$ (site I) moderately forms pre-SCs, whereas ABCE1$^{E485A}$ (site II) efficiently binds to 70S ribosomes in the presence of AMP-PNP and ADP and partially splits them even at non-permissive conditions. We conclude that nucleotide occlusion in the control site II triggers a conformation of ABCE1 primed to form a pre-SC. Notably, pre-SCs with the ATPase-deficient double mutant

ABCE1$^{E238A/E485A}$ could not be isolated as it splits most 70S ribosomes (Fig 2B). The essential role of site II in pre-SC formation is further accentuated by comparing the disengagement SR mutants (Fig 2C). If nucleotide occlusion in control site II is impeded by the S214R substitution, pre-SCs cannot form even with AMP-PNP. In contrast, ABCE1$^{S461R}$ displays a similar behavior to the WT. Thus, blocking nucleotide occlusion in the high-turnover site I does not affect pre-SC formation. To emphasize the importance of control site II for pre-SC formation, we created a mixed mutant S461R/E485A, which combines both strategies. Hence, site I is unable to close upon ATP binding, and site II is in an ATP-occluded state, deficient in ATP hydrolysis. In accordance, ABCE1$^{S461R/E485A}$ binds to 70S ribosomes with AMP-PNP and ADP (Fig 2D). Given that no additional occlusion event in site I can take place in the mixed mutant S461R/E485A, this corroborates our findings that ATP occlusion in site II induced by the catalytic E485A substitution is sufficient for pre-SC formation. Consistently, the reciprocal mutant S214R/E238A did neither bind nor split ribosomes and was consequently excluded from further investigations in downstream events of ribosome recycling.

### ATP occlusion in both sites drives ribosome splitting

After formation of the pre-SC, which requires nucleotide occlusion in site II of ABCE1, the ribosome is destined to be split apart. However, the contribution of each site to the potential power stroke remains opaque. We therefore monitored this step under single-turnover conditions with isolated components using purified *T. celer* 70S ribosomes at 25 mM Mg$^{2+}$ (Endoh et al, 2006, 2008; Becker et al, 2012) and at 45°C (Fig 3A). Ribosome splitting by ABCE1 was assisted by aRF1, and the addition of surplus aIF6 prevented reassociation of ribosomal subunits after a single round of splitting (Benelli et al, 2009). Under single-turnover conditions, WT ABCE1 splits ribosomes most efficiently with AMP-PNP in an aRF1-dependent manner (Fig 3B). No splitting was observed with ADP or in the absence of ABCE1 or aRF1. Strikingly, the ATPase of WT ABCE1 was accelerated sevenfold at splitting conditions in the presence of 70S and aRF1 (Figs 3 and S5), consistent with the specific activation of ABCE1 by assembled ribosomes (Pisarev et al, 2010; Shoemaker & Green, 2011). Ribosomes are also split by ABCE1 and aPelota (Figs 3 and S6). Although substitution of any catalytic glutamate in ABCE1 promoted ribosome splitting (Figs 3 and S7), it is surprising that even the substitution of both catalytic glutamates resulted in high splitting potential despite the significantly decreased ATPase activity of ABCE1$^{E238A/E485A}$. Thus, ribosome splitting per se does not directly depend on ATP hydrolysis (Fig 3D). Importantly, splitting is diminished with ADP in the case of ABCE1$^{E238A}$ (Fig 3D), highlighting the crucial role of ATP occlusion and subsequent structural changes in control site II. Hence, control site II triggers an intramolecular switch and activates the high-turnover site I in the free (Fig 1C) and ribosome-bound state (Figs 3 and S5). In turn, ATP occlusion and closure of site I drive the structural reorganization for ribosome splitting. Consistent with its function as molecular motor of ribosome splitting, we term the high-turnover site I a power-stroke site.

Along the ribosome recycling reaction, ABCE1 switches from a semi-closure of site II on the pre-SC to full closure of both sites on

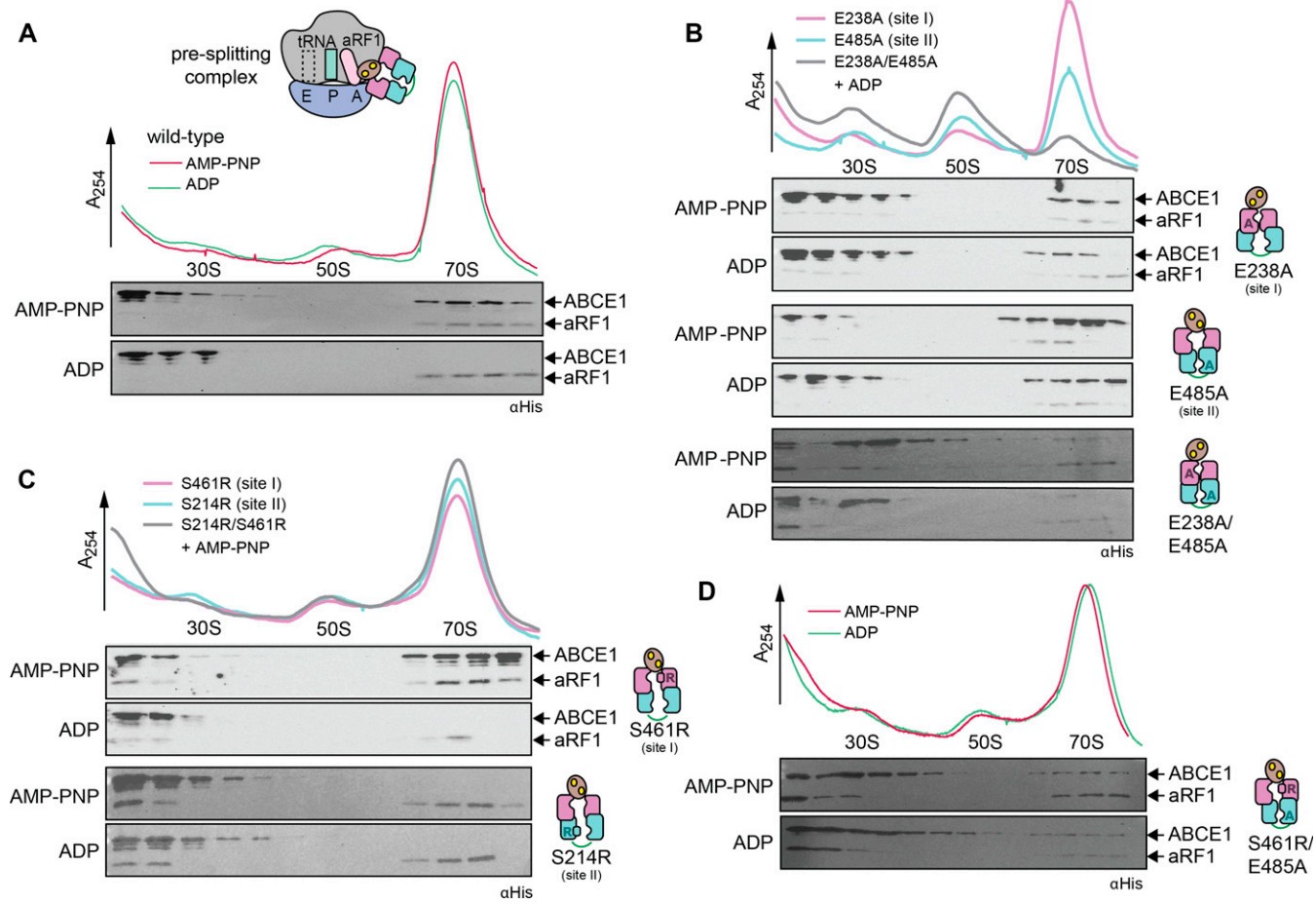

**Figure 2. Closure of site II is a prerequisite for pre-SC formation.**
**(A)** Pre-SC formation probed by SDG centrifugation. As expected, aRF1 is bound to the ribosome independently of the nucleotide supplemented, whereas WT ABCE1 is recruited only in the presence of AMP-PNP. **(B)** Exchange of any catalytic glutamate enables nucleotide-independent ribosome binding. Pre-SC formation could not be assayed for the double-EA variant as it splits ribosomes even at high $Mg^{2+}$ and low temperatures. Importantly, after splitting, ABCE1[E238A/E485A] remains bound at 30S as seen in the respective fractions of the immunoblot. **(C)** Steric hindrance by S214R mutation in site II prevents pre-SC formation even in the presence of AMP-PNP. In contrast, blocking site I by S461R mutation does not affect pre-SC assembly. **(D)** The mixed mutant S461R/E485A binds 70S in the presence of AMP-PNP and ADP. Representative set of two independent experiments.

the post-SC (Becker et al, 2012; Brown et al, 2015; Heuer et al, 2017). Consequently, ABCE1[E485A] or ABCE1[E238A/E485A] are primed to adopt the fully closed post-SC conformation and induce ribosome splitting with ATP, AMP-PNP, and ADP, whereas ABCE1[E238A] still requires ATP or AMP-PNP in the control site II to accomplish this task. Notably, a similar preference for the closed state has been reported for the catalytic base mutant of the homodimeric ABC transporter MsbA (Schultz et al, 2011). To disable the allosteric control of site I by site II, we analyzed the ribosome splitting ability of the disengagement mutants. None of them were able to split 70S, confirming that both sites must adopt a closed conformation to induce ribosome splitting (Fig 3E).

We independently studied the nucleotide occlusion in ABCE1 at splitting conditions. Mutations of one or both catalytic glutamates promote the stable binding of two nucleotides per ABCE1. Disengagement mutants with single substitutions in the ABC-signature motif, S214R or S461R, occlude only one nucleotide in the opposite, unmodified site. Background levels of nucleotide occlusion were

observed in the double SR mutant (Figs 4 and S8). WT ABCE1 was only partially occupied by ATP and ADP, consistent with the intermediate amount of split 70S compared with almost complete splitting by the EA variants (Figs 3C and S7). Thus, the exchange of one or both catalytic glutamates in ABCE1 facilitates the transition to a fully closed state with two occluded nucleotides, and therefore results in highly efficient ribosome splitting. The ATP-to-ADP ratios occluded by the single site mutants reflect the ATP turnover rate in the canonical site and illustrate the allosteric crosstalk between both asymmetric sites on single-turnover levels. ABCE1 with one catalytically active site harbors always one ATP molecule in the opposite ATPase-inactivated nucleotide-binding site, whereas the active site executes slow (E238A) or fast (E485A) ATP hydrolysis. As expected, the catalytically inhibited double E238A/E485A mutant occludes two unconverted ATP molecules. These single-turnover studies are consistent with the multiple (steady-state) ATPase activity assays (Figs 1C and D, and S2) and underline the significance of the low-turnover site II controlling the high-turnover site I.

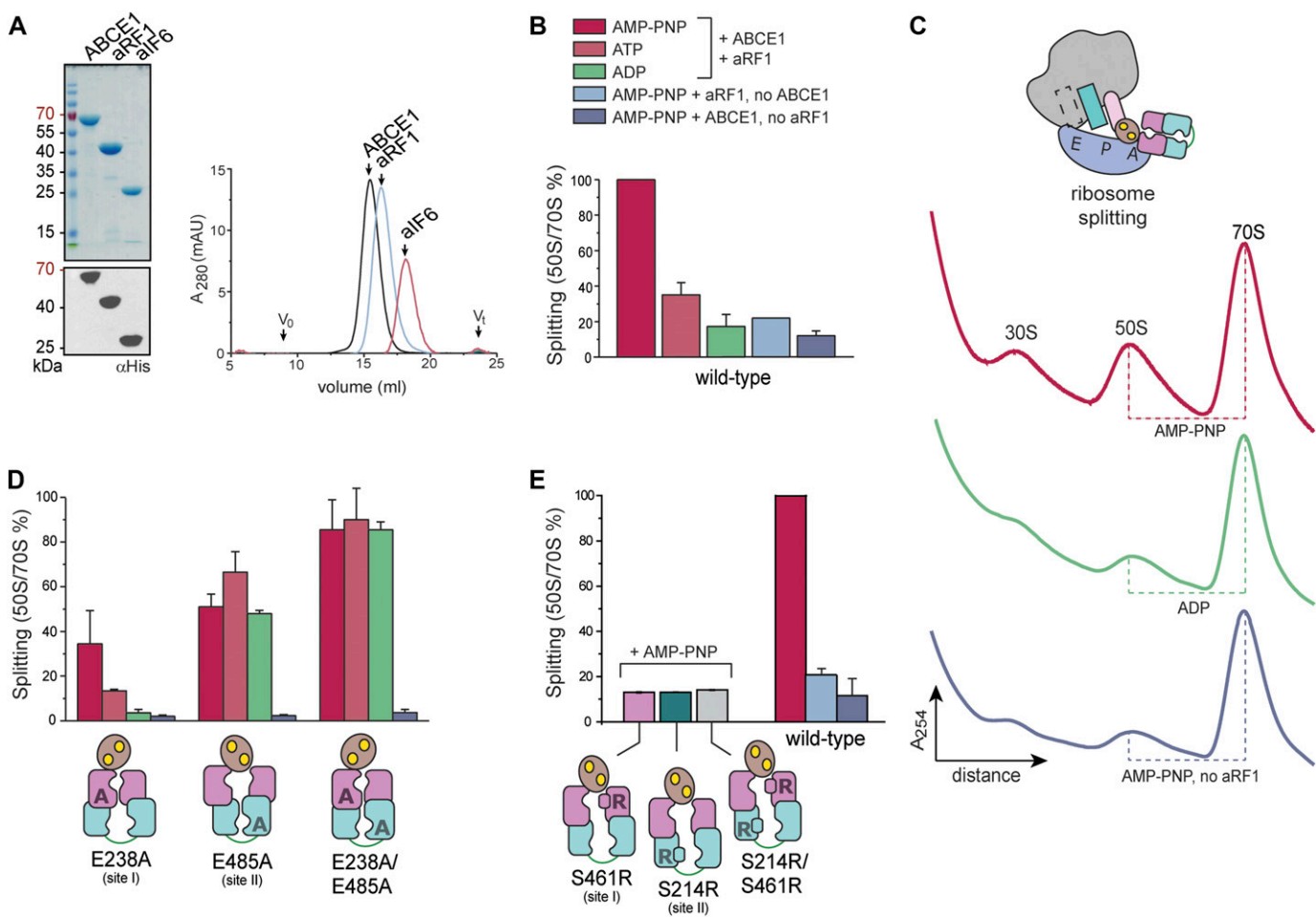

**Figure 3. Both nucleotide-binding sites must close for efficient 70S splitting.**
**(A)** A minimal set of splitting factors required for active splitting comprises ABCE1, aRF1, and aIF6 from *S. solfataricus*. All factors were pure and monodisperse as shown by SDS–PAGE (Coomassie), immunoblotting, and SEC. **(B)** Specific 70S splitting requires aRF1, ABCE1, and AMP-PNP. Results were normalized to the highest splitting ratio for WT ABCE1 with AMP-PNP and aRF1. **(C)** Traces corresponding to (B) clearly demonstrate an increase of 50S in the presence of WT ABCE1, aRF1, and AMP-PNP. **(D)** Ribosome splitting is most efficient, when both sites are in a closed, occluded state. Colors as in (B). **(E)** No ribosomes are actively split by any of the SR "disengagement" mutants, showing that closure of both sites is a prerequisite for ribosome splitting. Colors as in (B). Representative set of three independent experiments.

Altogether, these data demonstrate that the full closure of both sites, initiated by ATP occlusion in control site II, drives ribosome splitting.

### ATP hydrolysis in site II triggers the release of ABCE1 and terminates the post-SC

The formation of the post-SC was studied in cell lysates with purified ABCE1 variants. Efficient post-SC formation with WT ABCE1 requires AMP-PNP and elevated temperatures (Barthelme et al, 2011; Kiosze-Becker et al, 2016). The post-SC can neither be formed in the presence of ADP nor at 4°C (Fig 5A). Substitution of the catalytic glutamate in either site abolishes the requirement for AMP-PNP or elevated temperatures for post-SC formation. However, at low temperature, ABCE1$^{E485A}$ (site II) or the ABCE1$^{E238A/E485A}$ variant occupy the 30S ribosomal subunit significantly more efficient than ABCE1$^{E238A}$ (Figs 5 and S9). Furthermore, the S214R

mutation (preventing closure of control site II) eliminates post-SC formation even in the presence of AMP-PNP and high temperature. In contrast, the disengagement mutation S461R in the power-stroke site I does not impact post-SC formation (Fig 5B). To strengthen this conclusion, we analyzed post-SC formation by the mixed mutant ABCE1$^{S461R/E485A}$, which excludes a nucleotide occlusion event in site I. The additional disengagement mutation in site I (S461R) did not alter post-SC formation in comparison to E485A alone because binding was observed at 4°C with AMP-PNP and ADP. Hence, primarily, closure of control site II stabilizes the post-SC, consistent with the superior role of an intact site II in yeast viability studies (Figs 1E and S3). Mutants of E493 in site II block the formation of 80S by arresting ABCE1 at the post-SC after splitting, thus leading to cell death. In contrast, substitutions of the corresponding residue E247 in site I do not result in stable post-SCs and allow the assembly of translation-competent ribosomes without affecting cell viability. The ATPase activity of WT and mutant ABCE1 was strongly inhibited upon 30S binding (Figs 5C and D, and S9 and S10). Hence, after

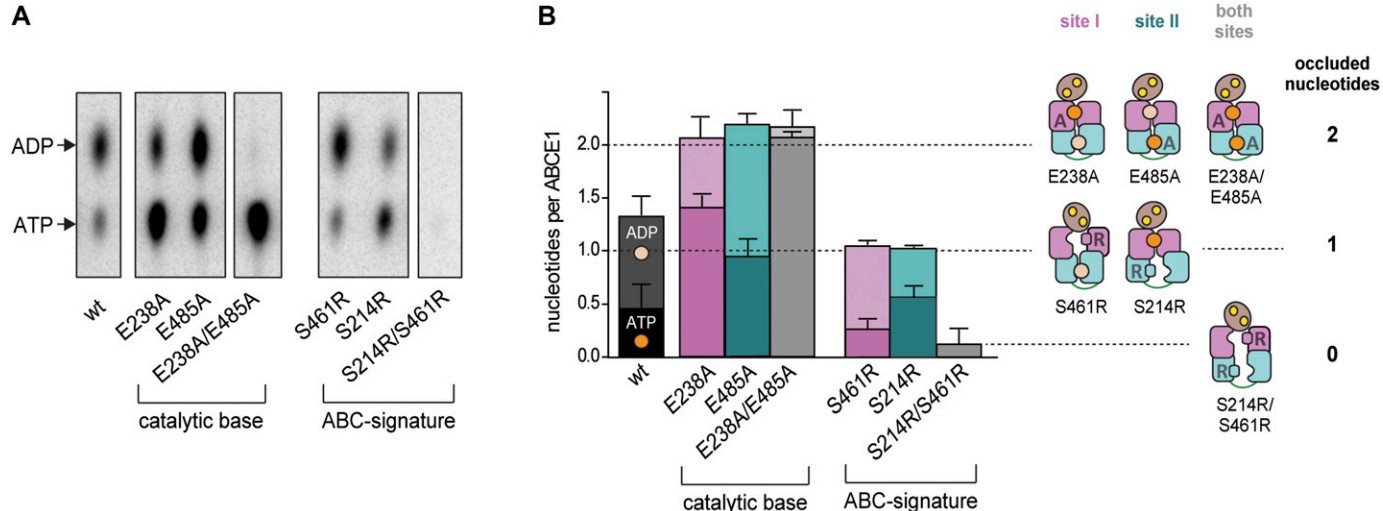

**Figure 4. ABCE1 occludes two nucleotides during 70S splitting.**
**(A)** Nucleotide occlusion is assayed at 70S splitting conditions by rapid gel filtration. Autoradiogram of the elution fractions from the nucleotide occlusion assay containing ABCE1 with the respective nucleotides securely trapped within the closed sites. Representative set of two independent experiments. **(B)** Exchange of the catalytic glutamates facilitates closure of the nucleotide-binding sites and ATP occlusion, hence, all EA variants occlude two nucleotides. As intended, introduction of arginine into the ABC-signature motif prevents nucleotide occlusion in the respective site, leading to one or a background of 0.2 nucleotides per protein for single-SR and double-SR substitutions, respectively. The ATP-to-ADP ratio occluded by the SR variants reflects the ATP turnover rate in the intact site.

ribosome splitting, ABCE1 remains bound to 30S in a fully closed state occluding two nucleotides until release by a yet undefined trigger.

## Discussion

Ribosome recycling is a fundamental process in cell homeostasis, growth, and differentiation, which must be regulated as tightly as translation initiation, elongation, and termination. We dissect the ribosome recycling event into discrete steps, each under control of the ribosome recycling factor ABCE1. Strategic mutations in ABCE1 reveal the functional asymmetry of the two nucleotide-binding sites in ATP hydrolysis, ribosome binding, and ribosome splitting. Based on our findings, we derive an elaborated working model of ribosome recycling catalyzed by ABCE1, which represents a novel ABC-type mechanism (Fig 6). Pre-SCs are formed when ABCE1 is recruited to ribosomes during classical termination or mRNA surveillance. In phase 1, the low-turnover site II adopts a semi-closed conformation upon ATP binding, acting as a checkpoint site in a modality similar to GTPases. In phase 2, an allosteric switch activates the high-turnover site I. The ATPase activity of ABCE1 is stimulated by splitting competent ribosomes and aRF1 but inhibited by the small ribosomal subunit. Thus, the power-stroke site I can hydrolyze several nucleotides in an attempt to occlude one ATP and switch to the closed conformation. Consequently, ABCE1 adopts a fully closed state with two occluded ATP and splits the ribosome. Full closure of both sites displaces the FeS cluster domain, which is allosterically coupled to site I (Heuer et al, 2017). The FeS cluster domain protrudes into the intersubunit space and causes a rearrangement in the pre-SC, thus leading to its destabilization and disassembly (Kiosze-Becker et al, 2016; Heuer et al,

2017). In phase 3, both sites remain locked in the closed form at the small ribosomal subunit. Consistently, ATP hydrolysis by ABCE1 is strongly inhibited within the post-SC. In phase 4 and 5, ATP hydrolysis schedules dissociation of ABCE1 from the post-SC, potentially triggered by initiation factors.

Each phase includes important checkpoints that regulate the progression of ribosome recycling. The occlusion of one ATP is essential for recognition of the post-TC with e/aRF1 or homologous factors. This evidence accounts for the regulation of translation in accord with the energy status of the cell. In addition, our model includes several ATP hydrolysis rounds in site I of 70S-bound ABCE1 at phase 2, which explains the previously observed ATP dependency of ribosome splitting (Pisarev et al, 2010; Shoemaker & Green, 2011) and represents an important checkpoint for ABCE1. Once engaged in a pre-SC with ATP occluded in control site II, ABCE1 can either occlude an additional ATP in power-stroke site I, close both sites, and split a terminated ribosome in an authorized recycling process; or hydrolyze ATP in site II, open, and dissociate from a splitting incompetent ribosome. Strikingly, phase 2 (splitting) and phase 3 (post-SC) explain the unequal impact of various site I and site II mutants in ABCE1 on cell viability (Dong et al, 2004; Karcher et al, 2005) and embryonic development (Coelho et al, 2005). Mutations interfering with closure of the nucleotide-binding sites (S223R and S469R), and thus preventing ribosome splitting in phase 2, are lethal but not dominant negative. These results are in accordance with a different set of ABC-signature mutants in yeast (G224D and G225D in site II, and G470D and G471D in site I) (Dong et al, 2004). In *D. melanogaster*, a mutation in the ABC-signature motif of site II (Q231L in LSGGEL<u>Q</u>) resulted in embryonic lethality (Coelho et al, 2005). In line with the proposed mechanism (Fig 6), the ABC-signature mutants of ABCE1 fail to split ribosomes (Fig 3) but are not permanently engaged in ribosomal complexes (Fig 5) and thus

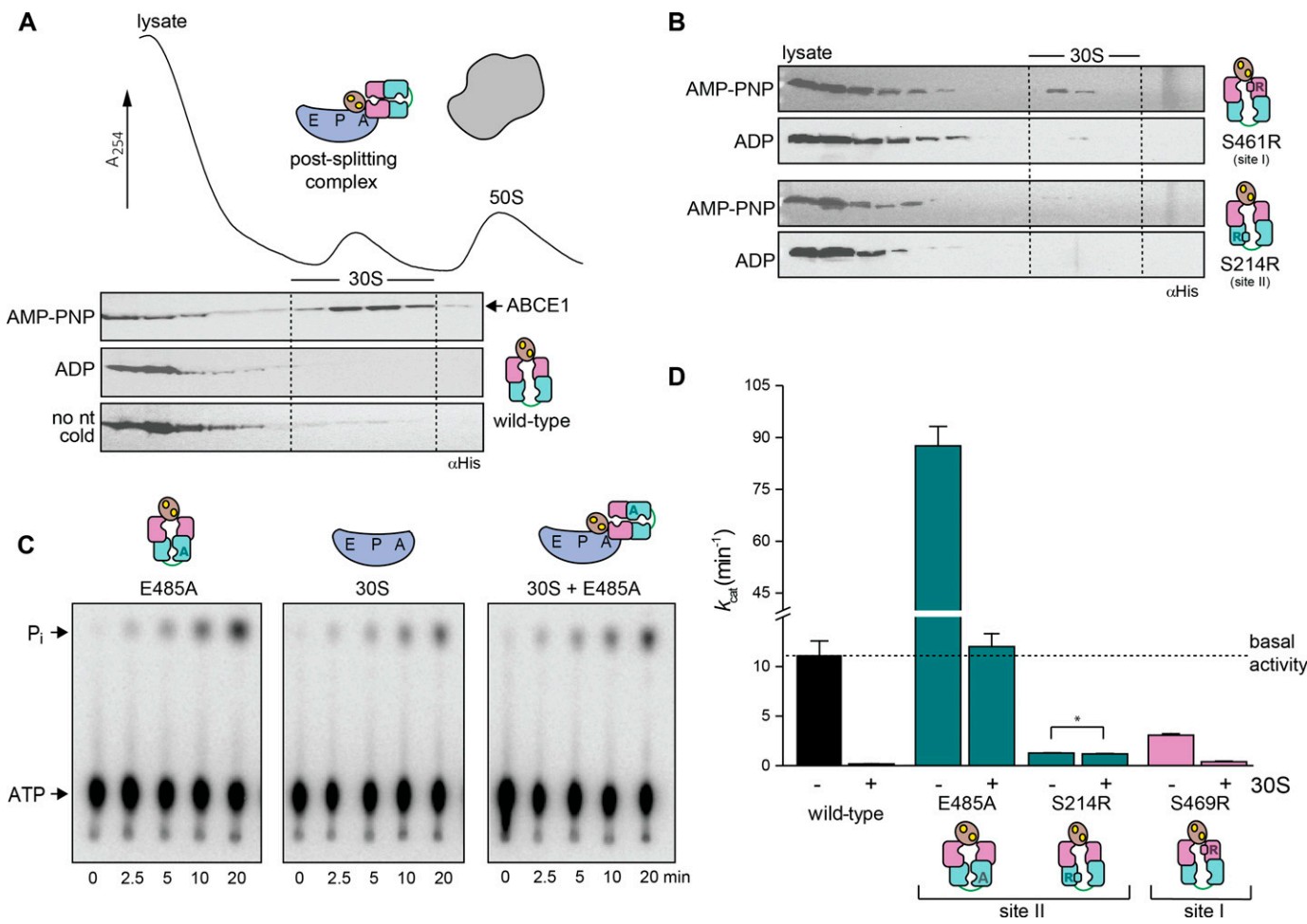

**Figure 5. Post-SC requires closure of site II and inhibits the ATPase activity of ABCE1.**
**(A)** The post-SC is assembled from *S. solfataricus* cell lysate (contains only 30S and 50S subunits) and recombinant ABCE1. WT protein essentially requires high temperature and AMP-PNP for 30S binding. **(B)** Blockage of site II by the S214R mutation severely inhibits post-SC formation. **(C)** 30S binding inhibits the ATPase activity of ABCE1$^{E485A}$ (1 µM), as demonstrated by TLC of $^{32}$P-γ-ATP (2 mM). ATP hydrolysis drops to the level of background 30S activity if the small subunit is added to the hyperactive E485A variant. **(D)** ATPase activity of ABCE1 is inhibited if a post-SC is efficiently formed. Strikingly, ATP hydrolysis rate of ABCE1$^{S214R}$ does not change upon addition of 30S (*) because the S214R mutation prevents 30S binding. The overall drop of $k_{cat}$ for ABCE1 in this experiment results from the higher Mg$^{2+}$ (20 mM) concentration used for 30S binding compared with 2.5 mM Mg$^{2+}$ in the ATPase measurements with ABCE1 only. Representative set of three independent experiments.

do not interfere with translation initiation on newly synthesized ribosomal subunits. In contrast, mutations preventing ATP hydrolysis and stabilizing the ATP occluded state in site II (E493A/Q) are dominant negative and lethal (Figs 1E and S3), whereas equal distractions in site I (E247A/Q) are tolerated, emphasizing the crucial control task of site II. Corresponding mutations are lethal in yeast (E247Q and E493Q) and fruit-fly (E501Q) (Dong et al, 2004; Karcher et al, 2005; Andersen & Leevers, 2007). The dominant negative effect of ABCE1$^{E493Q}$ (Dong et al, 2004) can now be explained by prolonged engagement of small ribosomal subunits in the post-SC (Fig S9) at phase 3 of ribosome recycling (Fig 6). This is consistent with decreased polysome levels and an inhibition of luciferase expression in whole-cell extracts (Dong et al, 2004). The analogous *D. melanogaster* PIXIE mutant E501Q shows a redistribution to 40S subunits (Andersen & Leevers, 2007).

Our proposed model for sequential ATP binding and hydrolysis in the active sites of ABCE1 during ribosome recycling is endorsed

by similar results for ABC transporters (Abele & Tampé, 2004). It supports the processive clamp or switch model for ABC proteins as simultaneous closure of both sites is required to split the ribosome, and their concurrent opening allows dissociation of ABCE1 from the small subunit. Allosteric regulation as in ABCE1 has been reported for other ABC-type proteins and involves crosstalk of the conserved D-loops (Grossmann et al, 2014; Hohl et al, 2014; Vedovato et al, 2015; Timachi et al, 2017). Strikingly, a division of work between two asymmetric nucleotide-binding sites has recently been reported for the gating cycle of the medically relevant ABC-transporter CFTR (Sorum et al, 2017).

How are ATP hydrolysis and subsequent ABCE1 release triggered on the post-SC at phase 4? Initiation factors may well serve this purpose regarding the previously demonstrated role of ABCE1 in translation initiation complex formation in yeast (Dong et al, 2004; Heuer et al, 2017) and humans (Chen et al, 2006). Structural data indicate a function of ABCE1 in translation initiation (Heuer et al,

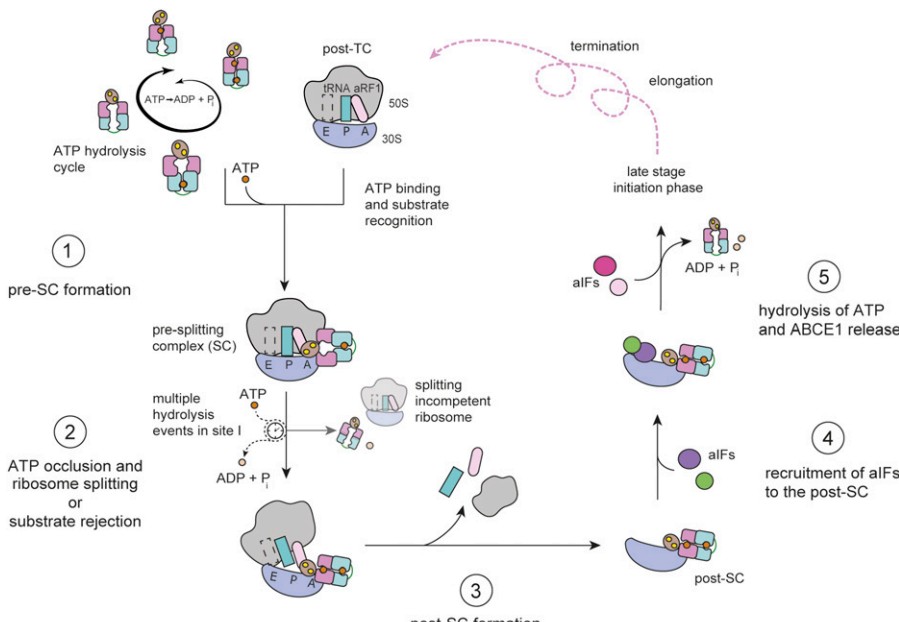

**Figure 6.   Molecular mechanism of ribosome recycling by ABCE1.**
Ribosome recycling is initialized by formation of the pre-SC via ABCE1 binding to assembled ribosomes. Substrate (post-TC) recognition is most efficient after binding of ATP in site II (see Fig 2). Pre-SC harbors ABCE1 with half-closed site II and open site I (step 1). In this conformation, site I is allosterically activated and can pass multiple hydrolysis rounds before one ATP is securely occluded and site I can close, which, in turn, leads to ribosome splitting as a second step in the recycling process (Figs 3 and 4). Alternatively, a splitting-incompetent post-TC is rejected after ATP hydrolysis in the control site II (step 2). A stable post-SC is formed with two closed nucleotide binding sites, significant for the third step of ribosome recycling. Post-SC formation is only possible if site II is occupied but, unlike the previous 70S splitting step, does not depend on the closure of site I (Fig 5; step 3). The fourth step connects ribosome recycling with translation initiation on the 30S subunit and includes recruitment of initiation factors in the presence of bound ABCE1 (step 4) as shown in recent cryo-EM reconstructions and early biochemical studies. The last step requires a trigger for ATP hydrolysis, which might be an external signal from the 30S subunit or a component of the initiation complex. Once both sites are open, ABCE1 dissociates from free or decorated 30S (step 5).

2017), thus supporting our early hypothesis of this versatile protein being the missing link between ribosome recycling and translation initiation (Nürenberg & Tampé, 2013). We suggest that the ribosome recycling factor ABCE1 acts as a regulator of mRNA translation and surveillance. In *Caenorhabditis elegans*, ABCE1 depletion results in embryonic lethality and slow growth (Zhao et al, 2004). ABCE1 depletion in *Xenopus laevis* fertilized eggs inhibited embryonic development before the late gastrula phase (Chen et al, 2006), thus further emphasizing the importance of ABCE1 for cell viability and embryonic development.

# Materials and Methods

### Cloning and site-directed mutagenesis

ABCE1, aRF1, and aIF6, cloned into pSA4 originated from pET15b with *amp* resistance, T7 promotor system, and C-terminal His$_6$-tag, were used. Point mutations were introduced by two-step megaprimer PCR as described (Barik, 1996).

### Protein expression

All proteins were expressed in *Escherichia coli* BL21 (DE3), co-transformed with *pRARE* plasmid (Novagen). Cells were grown overnight in LB medium with 100 μg/ml carbenicillin and 25 μg/ml chloramphenicol at 37°C and used to inoculate the main culture in TB medium with the same resistance markers at a ratio of 1:20. Cells were grown at 37°C until an $OD_{600}$ of 0.8, temperature was set to 20°C (aRF1, aPelota, and ABCE1) or 18°C (aIF6), and expression was induced by adding 1, 0.5, or 0.3 mM of IPTG for ABCE1, aRF1, or aIF6, respectively. Cells were harvested after 20–24 h (ABCE1 and aRF1) or 12 h (aIF6).

### Protein purification

All proteins were purified by using differential precipitation and two sequential chromatography steps (Fig S1), immobilized metal affinity chromatography (IMAC, HiTrap Chelating HP, 5 ml; GE Healthcare) and anion exchange chromatography (AIEX, HiTrap Q column, 1 ml; GE Healthcare) at room temperature. The Frozen cell pellet was supplemented 1:1 (vol/vol) with lysis-G buffer (20 mM Tris–HCl, pH 8.0, 300 mM NaCl, 5 mM MgCl$_2$, 40% glycerol [vol/vol], and 8 mM β-ME), and then thawed on ice. Cells were disrupted with five to eight pulses of 2 min on ice, using a Branson Sonifier 250 at 70% output. The lysate was centrifuged at 130,000 *g* for 20 min. The supernatant was incubated at 65°C for 10 min to precipitate host proteins, followed by a second centrifugation step at 130,000 *g* for 30–60 min. The supernatant of the second centrifugation step was used for purification of all archaeal proteins.

ABCE1 was purified by using an Äkta Express System (GE Healthcare) to minimize the contact time of the sensitive FeS clusters with air. All buffers were extensively degassed before purification. After loading, the IMAC column was washed with buffer IMAC-G$^{100}$ A (20 mM Tris–HCl, pH 8.0, 100 mM NaCl, 20 mM imidazole, 15% glycerol, and 2 mM β-ME) until the baseline of absorbance at 280 nm was reached. ABCE1 was then eluted by using 100% buffer IMAC-G$^{100}$ B (20 mM Tris–HCl, pH 8.0, 100 mM NaCl, 200 mM imid-azole, 15% glycerol, and 2 mM β-ME). The buffer was exchanged against AIEX-G$^{ABCE1}$ A (20 mM Tris–HCl, pH 8.5, 5 mM NaCl, 15% glycerol, and 2 mM β-ME) by using a Sephadex G-25 desalting column (GE Healthcare), and ABCE1 was further purified by using AIEX. After loading, the AIEX column was washed with buffer AIEX-G$^{ABCE1}$ A until the baseline was reached. ABCE1 was then eluted by using a gradient of 0–30% AIEX-G$^{ABCE1}$ B (20 mM Tris–HCl, pH 8.5, 1 M NaCl, 15% glycerol, and 2 mM β-ME). Fractions containing ABCE1 were identified by SDS–PAGE and the brown color of the FeS

clusters. Buffer of the pooled fractions was exchanged against Storage-G$^{150}$ buffer (20 mM Tris–HCl, pH 7.5, 150 mM NaCl, 15% [vol/vol] glycerol, and 2 mM β-ME) by using PD10 gravity flow desalting columns (Bio-Rad). Protein was concentrated using an Amicon Ultra centrifuge device (30 kD cut-off; Merck Millipore), snap-frozen in liquid nitrogen, and stored in small aliquots at −80°C. The protein concentration was determined at A$_{280}$ (ε$_{280}$ = 58,000 M$^{-1}$·cm$^{-1}$).

aRF1 and aPelota were purified by using an Äkta Prime System (GE Healthcare) using IMAC-G$^{240}$ buffers A (20 mM Tris–HCl, pH 8.0, 240 mM NaCl, 20 mM imidazole, 15% glycerol, and 4 mM β-ME) and B (20 mM Tris–HCl, pH 8.0, 240 mM NaCl, 200 mM imidazole, 15% glycerol, and 4 mM β-ME). After loading, the IMAC column was washed with IMAC-G$^{240}$ A until the baseline of absorbance at 280 nm was reached and aRF1/aPelota was eluted by using a short gradient (0–100% buffer B in 30 ml), which yielded one major peak that mostly contained a protein of the expected size as verified by SDS–PAGE. For subsequent AIEX, the buffer of all major peak fractions was exchanged to AIEX-G$^{aRF}$ A (20 mM Tris–HCl, pH 8.5, 40 mM NaCl, 4 mM MgCl$_2$ 15% glycerol, and 4 mM β-ME) by using PD10 gravity flow desalting columns (Bio-Rad). aRF1/aPelota was eluted from the AIEX column by using a flat gradient; 0–30% AIEX-G$^{aRF}$ B (20 mM Tris–HCl, pH 8.5, 1 M NaCl, 4 mM MgCl$_2$ 15% glycerol, and 4 mM β-ME) in 60 ml. Fractions containing aRF1/aPelota were identified by SDS–PAGE. The buffer of the pooled fractions was exchanged against Storage-G$^{250}$ buffer (20 mM Tris–HCl, pH 7.5, 250 mM NaCl, 5 mM MgCl$_2$, and 15% glycerol) by using PD10 gravity flow desalting columns (Bio-Rad). The protein was concentrated using the Amicon Ultra centrifuge device (10 kD cut-off; Merck Millipore), snap-frozen in liquid nitrogen, and stored in small aliquots at −80°C. Protein concentration was determined at A$_{280}$ (ε$_{280}$ = 35,000 M$^{-1}$·cm$^{-1}$).

aIF6 was purified by using the Äkta Prime system (GE Healthcare) using IMAC-G$^{300}$ buffers A (20 mM Tris–HCl, pH 8.0, 300 mM NaCl, 20 mM imidazole, 15% glycerol, and 2 mM β-ME) and B (20 mM Tris–HCl, pH 8.0, 300 mM NaCl, 200 mM imidazole, 15% glycerol, and 2 mM β-ME). After loading, the IMAC column was washed with IMAC-G$^{300}$ until the baseline was reached, followed by an additional washing step with three column volumes 20% IMAC-G$^{300}$ B before elution with 100% IMAC-G$^{300}$ B. Pooled fractions were dialyzed against AIEX-G$^{aIF6}$ A (20 mM Tris–HCl, pH 8.5, 5 mM NaCl, 1 mM MgCl$_2$, 15% glycerol [vol/vol], and 2 mM β-ME) overnight at 4°C in a dialysis cassette (7 kD cut-off, Slide-A-Lyzer; Thermo Fisher Scientific) and loaded onto the equilibrated AIEX column. aIF6 was eluted from the AIEX column by using a flat gradient of 0–30% AIEX-G$^{aIF6}$ B (20 mM Tris–HCl, pH 8.5, 1 M NaCl, 1 mM MgCl$_2$, 15% glycerol, and 2 mM β-ME) in 60 ml. Fractions containing aIF6 were identified by SDS–PAGE. The buffer of the pooled fractions was exchanged against Storage-G$^{300}$ buffer (20 mM Tris–HCl, pH 7.5, 300 mM NaCl, 5 mM MgCl$_2$, and 15% glycerol) by using PD10 gravity flow desalting columns (Bio-Rad). The protein was concentrated using the Amicon Ultra centrifuge device (10 kD cut-off; Merck Millipore), snap-frozen in liquid nitrogen, and stored in small aliquots at −80°C. The protein concentration was determined at A$_{280}$ (ε$_{280}$ = 5,700 M$^{-1}$·cm$^{-1}$).

### Analytical size exclusion chromatography (SEC)

ABCE1, aRF1, aPelota, and aIF6 were analyzed by using analytical SEC (Superdex 200, 24 ml; GE Healthcare) at room temperature in Storage-G$^{250}$ buffer. All variants of ABCE1 were additionally run on a 2.4-ml Superose 6 (GE Healthcare) on an Äkta Ettan Chromatography System (GE Healthcare) at 4°C in Storage-G$^{150}$ buffer without glycerol recording absorption at 280 and 410 nm to analyze the integrity of the FeS clusters (Fig S1).

### Purification of *T. celer* 70S ribosomes

Frozen cell pellets from *T. celer* were purchased from the Centre of Microbiology and Archaea, University of Regensburg, Germany. Ribosomes were purified according to Becker et al, (2012). Cells were resuspended in S30 buffer (10 mM Tris–HCl, pH 7.5, 60 mM KOAc, 14 mM MgCl$_2$, and 1 mM DTT) and lysed by ultra-sonication with two to three rounds of 1 min on ice using a Branson Sonifier 250 at 60% output. Supernatant was cleared twice by centrifugation for 30 min at 30,000 $g$ at 4°C. Ribosomes were stripped from all translation factors by a high-salt cushion (1 M sucrose, 0.5 M NH$_4$OAc, and S30 buffer) during centrifugation for 1 h at 100,000 $g$ at 4°C. The high-salt cushion was removed and the pellet was resuspended in TrB25 (56 mM Tris–HCl, pH 8.0, 250 mM KOAc, 80 mM NH$_4$OAc, 25 mM MgCl$_2$, and 1 mM DTT). The absorption was measured at 280 and 260 nm for control. Pure 70S ribosomes were obtained by SDG centrifugation on a 10–30% (wt/vol) sucrose gradient in 10 mM Tris–HCl, pH 7.5, 60 mM KOAc, and 20 mM MgCl$_2$ at 20,000 rpm in an SW41 rotor (Beckmann Coulter) at 4°C for 14 h. Gradients were harvested by using a Piston Gradient Fractionator (BioComp Instruments) recording absorption at 254 nm (Bio-Rad). Fractions containing only 70S were concentrated using an Amicon Ultra centrifuge device (100 kD cut-off; Merck Millipore), snap-frozen in liquid nitrogen, and stored in small aliquots at −80°C. The ribosome concentration was estimated at A$_{260}$ (ε$_{260}$ = 4.2 × 10$^7$ M$^{-1}$·cm$^{-1}$).

### ATPase assay

ATPase activity of ABCE1 was measured by hydrolysis of $^{32}$P-γ-ATP (222 TBq/mmol, 370 MBq/ml; Hartmann Analytics) and subsequent TLC on polyethylene imine plates (Merck Millipore) using a 0.8 M LiCl solution in 0.8 M acetic acid (Pisarev et al, 2010). 10-fold cold ATP were supplemented 1:1,000 with radioactive tracer. A final concentration of 1 µM ABCE1 and 5 mM ATP in 20 mM Hepes, pH 7.5, 150 mM NaCl, 2.5 mM MgCl$_2$, and 1 mM DTT was used in a total volume of 50 µl for measurements with free ABCE1 at 70°C. For ATPase stimulation 1 µM *T. celer* 70S, 1 µM ABCE1, and 37.5 µM ATP in TrB25 at 45°C were used. For post-SCs, 4 µM *S. solfataricus* 30S, 1 µM ABCE1, and 2 mM ATP in RB buffer (20 mM Hepes-KOH, pH 7.5, 100 mM KCl, 20 mM MgCl$_2$, and 2 mM DTT) at 65°C were used. Spots were set by withdrawing a 1-µl sample at each point in time. After separation of the compounds, the plates were dried and exposed to a radio screen (Bio-Rad) overnight. Spots were quantified using ImageJ (NIH), and data were analyzed using Origin (OriginLab). The values of ATP auto-hydrolysis in samples without ABCE1 were subtracted during analysis. Free ABCE1 was measured three times. Bars in Fig 1C show the mean ± SD of a representative time–course experiment. Time–course measurements with 70S were performed twice. Bars in Fig S5 represent the mean ± SD. Time–course measurements with 30S were performed three times. Bars in Figs 5 and S9C represent

the mean ± SD. Radiograms in Fig S10 are a representative set of three independent experiments.

### Plasmid shuffling assay

Viability of ABCE1 mutants was checked as previously described (Heuer et al, 2017). The haploid yeast strain CEN.MG1-9B (*MATa his3Δ1 leu2-3,112 trp1-289 MAL2-8^C SUC2 ura3-52 rli1::KanMX4* + pRS426-ABCE1) was generated in which the essential *ABCE1* gene (*RLI1*) was deleted by *KanMX4* and substituted by pRS426-ABCE1 expressing WT *ABCE1* under the control of the endogenous pro-moter. The CEN.MG1-9B strain was transformed with pRS423-ABCE1 [*HIS*] plasmid coding for WT and mutated *ABCE1* and with pRS423 as negative control and selected on −HIS. The strain was prone to survive only in the presence of pRS423-ABCE1 by selection on −HIS and 5-FOA that activates the toxic activity of the pRS426-ABCE1 [*URA*] plasmid. Growth and survival were checked by growth studies in a serial dilution assay over 14 h. Data in Figs 1E and S3 are representative of a set of two independent experiments.

### 70S ribosome binding assay

Formation of the pre-SC was analyzed by SDG centrifugation, subsequent fractionation, protein precipitation, and immuno-blotting as described previously (Barthelme et al, 2011; Becker et al, 2012). Samples of 25 μl in TrB50 (as TrB25 but with 50 mM MgCl₂) contained 125 pmol 70S, 150 pmol ABCE1, 125 pmol aRF1, and 2 mM nucleotides and were incubated at 25°C for 1 h. Samples were cooled down on ice, loaded onto a 10–30% (wt/vol) SDG in TrB50, and centrifuged at 40,000 rpm in an SW41 rotor for 3 h. Gradients were fractionated by using a Piston Gradient Fractionator (BioComp Instruments) into 0.5 ml fractions. Those were precipitated by addition of ice-cold acetone overnight and pelleted by centrifu-gation at 16,000 *g* for 1 h; the pellets were resuspended in ATPase buffer before analysis by SDS–PAGE and immunoblotting. The immunoblots are representative of two independent experiments for each mutant.

### Ribosome splitting assay

70S splitting was analyzed by SDG centrifugation and subsequent absorption read-out at 254 nm. For the reaction, 25 pmol of 70S, 100 pmol of ABCE1 and aRF1, 180 pmol of aIF6, and 2 of mM nucleotides were incubated for 25 min at 45°C in a total volume of 50 μl in TrB25. The reaction was stopped by rapid cooling on ice and loaded onto a 10–30% (wt/vol) SDG in TrB50. Gradients were centrifuged in an SW41 rotor (Beckmann Coulter) at 20,000 rpm for 14 h or 40,000 rpm for 3 h at 4°C, and data were recorded at 254 nm by using a Piston Gradient Fractionator (BioComp Instruments). Splitting experi-ments were performed three times; the bars represent a mean ± SD value of the 50S/70S peak height ratio (Figs 3C and S7) normalized to the mean value of WT ABCE1 (Fig 3B and E) or to the highest value, reached by ABCE1^E238A/E485A (Fig 3D). Depicted SDG profiles are always representative of all three independent experiments. Splitting assays comparing aRF1 with aPelota (Fig S6) were per-formed three times independently of previous experiments. Bars show a normalized mean ± SD.

### Nucleotide occlusion

The occlusion of ATP and ADP by ABCE1 was determined by using ³²P-α-ATP (222 TBq/mmol, 370 MBq/ml; Hartmann Analytics) and the analysis as already described for the ATPase assay. Here, 9 μM cold ATP was supplemented 1:500 with radioactive tracer, and final concentrations of 0.6 μM ATP and 0.3 μM ABCE1 were incubated for 30 s at 45°C in TrB25. Samples were then quickly chilled on ice and supplemented with 0.5 mM cold ATP to reduce unspecific binding. To determine the intensity of the load, a 1-μl sample was directly spotted onto the thin layer chromatography (TLC) plate. ABCE1 and occluded ATP molecules were separated from residual ATP by SEC in Micro Bio-Spin P30 columns (Bio-Rad). 1 μl of the eluted sample was used for TLC analysis. The signals for ATP and ADP in the load samples summed up to a total corresponding to 0.6 μM of ATP. Retention of ABCE1 by the SpinColumn was calculated using SDS–PAGE analysis. An example of the calculation procedure is given in Fig S8. Nucleotide occlusion was preformed twice. Bars in Fig 4B represent a mean ± SD value and the radiogram in Fig 4A is representative of both independent experiments.

### 30S binding assay

*S. solfataricus* lysate was used as the source of 30S ribosomal subunits. Lysate was prepared as for 70S purification from frozen cells grown as described previously (Barthelme et al, 2011). The lysate was diluted 1:1 with RB buffer, 0.5 μM of ABCE1, and 2 mM of nucleotides. The reaction proceeded at 65°C for 10 min. Samples were cooled down on ice and loaded onto 5–15% (wt/vol) SDG in 20 mM Tris–HCl, pH 7.5, 10 mM KCl, 20 mM MgCl₂, and 1 mM DTT. Gradients were centrifuged, fractionated, and further analyzed as for 70S binding. Immunoblots are representative of a set of three independent experiments for each mutant.

## Supplementary Information

## Acknowledgements

We thank Simon Trowitzsch, Kristin Kiosze-Becker, Umar Jan, Stefan Brüchert, and Bianca Hetzert for fruitful discussions. E Nürenberg-Goloub was supported by the Christiane Nüsslein-Volhard foundation, L'Oréal, and the United Nations Educational, Scientific and Cultural Organization. M Gerovac was supported by the Boehringer Ingelheim Fonds. The German Research Foundation (DFG) SFB 902 "Molecular mechanisms of RNA-based regulation" and the Cluster of Excellence EXC114 funded this work (to R Tampé).

### Author Contributions

E Nürenberg-Goloub: conceptualization, investigation, formal analysis, validation, visualization, methodology, writing—original draft, review, and editing.
H Heinemann: formal analysis, validation, and investigation.

M Gerovac: formal analysis, validation, investigation, and visualization.
R Tampé: conceptualization, resources, formal analysis, supervision, funding acquisition, investigation, methodology, writing—original draft, project administration, and writing—review and editing.

## Conflict of Interest Statement

The authors declare that they have no conflict of interest.

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
