## [Reviewer comments · Life Science Alliance]

Ribosome recycling is coordinated by processive events in two asymmetric ATP sites of ABCE1

Elina Nürenberg-Goloub, Holger Pfitzner, Milan Gerovac, and Robert Tampé

DOI: 10.26508/lsa.201800095

Review timeline:	Submission date:	27 May 2018
	1 st Revision received:	27 May 2018
	1 st Editorial Decision:	28 May 2018
	2 nd Revision received:	31 May 2018
	Accepted:	1 June 2018

Report:

(Note: Letters and reports are not edited. The original formatting of letters and referee reports may not be reflected in this compilation.)

Referee #1:

In this manuscript, the Tempé group analyzes the mechanism of ribosome splitting by the ABCE1 ATPase. Using mutants of the two ATP binding/hydrolyzing domains alone or in combination and ATPase assays, ribosome binding or ribosome dissociation assays, the authors propose a mechanism for archaeal/eukaryotic ribosome splitting.

Little being known on ribosome splitting by ABCE1 ATPases, a strength of this manuscript is to provide some insights into this mechanism. The results presented are based on high-level biochemical analyses. The model proposed by the authors rests, however, on an analysis of mutant factors. It would have been informative to provide independent lines of research to support the authors model. In this situation, while this manuscript provides interesting information that could be published in The EMBO Journal, it could also be considered for a shorter format such as the one of EMBO Reports.

Other points:

- Page 5, line 4: The sentence presenting similarity between archaeal, yeast and human ABCE1 is poorly written and difficult to read.
- Figure 1, panel E: In the left column spotting is not too heterogeneous (-His media). It is difficult draw conclusions from this experiment. Moreover, cells are too dilute, most spots being barely visible. This panel is thus not informative and this experiment should probably be repeated (more homogenous dilutions, less dilute starting samples...).
- The discussion is very long and duplicates part of the Results section. It could be refocused.
- Authors should perhaps compare their model of ribosome splitting with mechanisms occurring in bacteria.

Referee #2:

The twin-ATPase ABCE1 is the ribosome recycling factor in archaea and eukarya. Ribosome recycling is an essential phase of protein synthesis and splits the ribosome into subunits after translation termination. The overall pathway of ribosome recycling has been worked out by several previous functional and structural studies. ABCE1 binds to ribosomal post-termination 70S/80S complexes containing deacylated tRNA and a/eRF1. Splitting results in post-splitting complexes with ABCE1 still bound to the 30S/40S subunit. This may connect ribosome recycling to initiation, before ABCE1 is released. However, in order to understand the mechanism of ABCE1 with respect to order and timing of ATP binding, occlusion, and hydrolysis in the two nucleotide binding sites a thermodynamic and kinetic framework for ABCE1 action is required.

In the present, paper Nürenberg-Goloub et al. have introduced a set of point mutations at strategic residues of the two nucleotide-binding domains (NBDs) in *S. solfataricus* ABCE1. Based on previous studies these mutations in either site are thought to occlude ATP by preventing ATP or to prevent ATP occlusion by the introduction of bulky residues. The set of mutated ABCE1 was then tested for ATPase activity, ribosome binding and splitting activity. While the study is potentially interesting, there are several problems associated with the manuscript:

1. The effects of the mutations are only partially quantified. For ribosome binding the authors performed SDG centrifugation and estimated the amount of bound ABCE1 to 70S and 30S fractions by visual inspections of immunoblots. This is not adequate. Quantitative measurements and the estimation of binding constants is

desirable.

2. For ribosome splitting SGD centrifugations were performed. It is not described how exactly the 50S/70S ratio was derived. By height of the peaks or by integration of the peaks? The authors state on page 7 that "no splitting was observed with ADP or in the absence of ABCE1 or aRF1". This statement is in contrast to the data presented in Fig. 3B,C where 30S and 50S subunit peaks are clearly visible in the SGD centrifugation profiles and by the estimate of about 20% splitting. What is the reason for the background? On the other hand they state that "Wildtype ABCE1 splits ribosomes most efficiently with ATP or AMP-PNP in an aRF1 dependent manner." However, in the presence of ATP splitting is estimated to be less than 40% compared to the AMPPNP case. This is confusing.

3. The 50S/70S ratio as a measure for splitting (Fig. 3B) is given only normalized to the splitting with AMP-PNP, aRF1 and ABCE1. What is the absolute value? From Fig. 3 it seems that even in this 100% case the majority of the 70S ribosomes is not split. What is the reason? This requires explanation and discussion.

4. As written, the asymmetry of the two nucleotide binding sites and the hyperactivity of ABCE1(E485A) caused by ATP occlusion in site II is presented as a novel finding. The finding of two asymmetric ATP-binding sites is even put into the Highlights. However, 7 years ago the Tampe group has already suggested functional and structural asymmetry of the two ATP-binding based on ATPase hyperactivity caused by the related E485Q mutant (Barthelme et al., 2011, PNAS). These related findings are not properly discussed. Barthelme et al., 2011 is cited several times throughout the manuscript for more general statements, but in the section "Two nucleotide-binding sites operate asymmetrically in ABCE1" no reference to this related paper can be found. This is not right.

5. In Fig. 1C the *k*_{cat} for ATPase by wt ABCE1 is given as ~20/min. In Fig.3 extended view 1C a *k*_{cat} of 0.32/min is given for ABCE1 alone. What is the reason for this discrepancy?

6. The manuscript is very hard to read, especially for the non-expert reader. In general it may help to elaborate on how certain conclusions have been derived. Especially for the figure legends, it would be helpful, if the figure content is better described. The authors also should check the text for consistency. For example on page 7 they first state that "ABCE1 splits ribosomes most efficiently with ATP or AMP-PNP" and then wonder that "it is surprising that even the substitution of both catalytic glutamates resulted in high splitting potential by the ATPase inactive ABCE1, thus demonstrating that ATP hydrolysis per se is not required to drive ribosome splitting". Why is this surprising? Isn't this expected from efficient splitting in the presence of the non-hydrolysable AMP-PNP?

Referee #3:

In this study the interplay of the two individual ATP active sites of the ribosome recycling factor ABCE1 is detailed. By employing mutants that either block ATP-hydrolysis or ATP binding, the authors aim to detail the allosteric communication that takes place between the two domains. They begin by demonstrating the effect that blocking hydrolysis of either has on the turnover of the other. They see a dramatic enhancement of turnover when the catalytic residue of site II mutated. When paired with the observation that ATP binding mutants at site II lead to a clear decrease in ATP turnover, the authors conclude that ATP occupancy of this site serves a 'check-point' for the overall cycle.

They also show ATP turnover for double mutants of both the hydrolysis- and binding-residues. There is clearly more turnover in the hydrolysis double mutants, suggesting that residual activity still remains. No comment is made as to the limit of their detection. If this is genuine signal, than many of their further results are called into question, as ATP hydrolysis has not truly been abolished. The yeast results cannot be summarized due to figure quality.

Sucrose density gradients are utilized to show how nucleotide state influences formation of pre-splitting complexes. This passage is frustrating to follow, with many parenthetical notes and a seeming lack of consistency in nomenclature for both conditions assayed and how mutants are referenced. The constant shuffling of the order that mutants in the figures are presented makes interpretation especially difficult, this is exacerbated with no consistent depiction of nucleotide condition. This section manages to highlight the need to have ATP occupancy of site II in order to associate with the 70S ribosome.

The gradient profiles provided in this section are largely incomplete. For example the gradient for the double hydrolysis mutant in the presence of ADP is provided in 2B, yet the blot shown at the bottom is very poor quality. The blot in the presence of AMPPNP looks better, yet the gradient profile is not provided. No profiles are provided in the absence of nucleotide for any species as a control.

They go on to try and demonstrate the ability to split ribosomes is not a function of nucleotide hydrolysis, but of binding. Employing surplus IF6 prevents reassociation by binding the large subunit, allowing for the monitoring of a single round of splitting events. For WT ABCE1, splitting is optimal in the presence of AMPPNP and is diminished with ATP and ADP. It is confusing as to why there is such a drastic loss of splitting function for the two individual hydrolysis mutants in the presence of AMPPNP. One would expect that a nonhydrolyzable nucleotide would elicit the same splitting function regardless of the ability of a subunit to carry out nucleotide hydrolysis.

Does this suggest a contamination of ATP in the AMPPNP prep? Do the catalytic residues also influence affinity for nucleotide? They also state that "it is surprising that even the substitution of both catalytic glutamates resulted in high splitting potential by the ATPase inactive ABCE1." This cannot be said, as the supplement for figure 1 shows there is still hydrolysis.

The discussion around E485A and the double hydrolysis mutant is very confusing. They state that these species are "primed to adopt the fully closed conformation and induce ribosome splitting," with any nucleotide. Why? It is certainly clear that they are active under conditions where E238A is not, but without some more insightful piece of data this is something that needs to be omitted from the results.

The nucleotide binding and recovery assays are presented in figure 4. This is a population level measurement however all data is presented as a single molecule result. It is not at all clear how the authors arrive at plot of figure 4A from their provided method. It is understandable that when blocking hydrolysis you will maximally occupy the two binding sites of the protein, but how is that quantification actually being done in the experiment. Is this just an assumption that is being made? It is again asserted that the double hydrolysis mutant is catalytically inactive despite already demonstrating that it does indeed have activity. This assay is done at very different conditions than their previous ATPase experiments. Has part of panel B been cut off?

The data provided in figure 5 suggests that nucleotide occupancy in site II is required for formation of the post splitting complex. This assay is performed in lysate, presumably there are intact 70s ribosomes present? Wouldn't including the entire density gradient be informative for the way the different nucleotide states for each mutant alter the total distribution of ABCE1 with different ribosome species?

1st Revision – authors' response

27 May 2018

Reviewer #1:

In this manuscript, the Tampé group analyzes the mechanism of ribosome splitting by the ABCE1 ATPase. Using mutants of the two ATP binding/hydrolyzing domains alone or in combination and ATPase assays, ribosome binding or ribosome dissociation assays, the authors propose a mechanism for archaeal/eukaryotic ribosome splitting.

Little being known on ribosome splitting by ABCE1 ATPases, a strength of this manuscript is to provide some insights into this mechanism. The results presented are based on high-level biochemical analyses. The model proposed by the authors rests, however, on an analysis of mutant factors. It would have been informative to provide independent lines of research to support the authors model. **In this situation, while this manuscript provides interesting information that could be published in The EMBO Journal, it could also be considered for a shorter format such as the one of EMBO Reports.**

Other points: -Page 5, line 4: The sentence presenting similarity between archaeal, yeast and human ABCE1 is poorly written and difficult to read.

Reply: *Has been changed accordingly.*

-Figure 1, panel E: In the left column spotting is not too heterogeneous (-His media). It is difficult draw conclusions from this experiment. Moreover, cells are too dilute, most spots being barely visible. This panel is thus not informative and this experiment should probably be repeated (more homogenous dilutions, less dilute starting samples...).

Reply: *With our yeast viability assay, we aimed to also illustrate the dominant negative effects of site II catalytic base mutants of ABCE1 on yeast viability (left panel). To capture these results, yeast was grown for only 14 h (now in particularly stated in the Materials & Methods section), which unfortunately resulted in strongly diminished growth in the presence of 5-FOA (right panel). The contrast settings of this panel have been adjusted and the lower dilutions have been cut off to reduce the size of the panels.*

-The discussion is very long and duplicates part of the Results section. It could be refocused.

Reply: *We have removed redundancies from the results section. Since our results are related to both fields of ribosome recycling and ABC systems and since we couple them to previously published physiological effects in various organisms, shortening the discussion would reduce the impact of our manuscript.*

-Authors should perhaps compare their model of ribosome splitting with mechanisms occurring in bacteria.

Reply: *Bacteria have a totally different mode of translation termination and ribosome recycling, while archaeal and eukaryotic systems share most features. Hence, a comparison with bacteria would shift the manuscript out of focus.*

Reviewer #2:

The twin-ATPase ABCE1 is the ribosome recycling factor in archaea and eukarya. Ribosome recycling is an essential phase of protein synthesis and splits the ribosome into subunits after translation termination. The overall pathway of ribosome recycling has been worked out by several previous functional and structural studies. ABCE1 binds to ribosomal post-termination 70S/80S complexes containing deacylated tRNA and a/eRF1. Splitting results in post-splitting complexes with ABCE1 still bound to the 30S/40S subunit. This may connect ribosome recycling to initiation, before ABCE1 is released. However, in order to understand the mechanism of ABCE1 with respect to order and timing of ATP binding, occlusion, and hydrolysis in the two nucleotide binding sites a thermodynamic and kinetic framework for ABCE1 action is required.

In the present, paper Nürenberg-Goloub et al. have introduced a set of point mutations at strategic residues of the two nucleotide-binding domains (NBDs) in *S. solfataricus* ABCE1. Based on

previous studies these mutations in either site are thought to occlude ATP by preventing ATP or to prevent ATP occlusion by the introduction of bulky residues.

Reply: *We clearly show these effects by our ATP hydrolysis and ATP occlusion assays.*

The set of mutated ABCE1 was then tested for ATPase activity, ribosome binding and splitting activity. While the study is potentially

interesting, there are several problems associated with the manuscript:

1. The effects of the mutations are only partially quantified. For ribosome binding the authors performed SDG centrifugation and estimated the amount of bound ABCE1 to 70S and 30S fractions by visual inspections of immunoblots. This is not adequate. Quantitative measurements and the estimation of binding constants is desirable.

Reply: *For the determination of binding constants, a very large set of additional experiments and huge amounts of ribosomes are necessary, which we unfortunately cannot provide.*

2. For ribosome splitting SDG centrifugations were performed. It is not described how exactly the 50S/70S ratio was derived. By height of the peaks or by integration of the peaks?

Reply: *As correctly stated by the referee, ribosome splitting was quantified by the height ratio of the 50S and 70S peaks, which is now mentioned in Material & Methods. We further provide exemplary ratios for the SDG profiles presented in Figure 3 EV3.*

The authors state on page 7 that "no splitting was observed with ADP or in the absence of ABCE1 or aRF1". This statement is in contrast to the data presented in Fig. 3B,C where 30S and 50S subunit peaks are clearly visible in the SDG centrifugation profiles and by the estimate of about 20% splitting. What is the reason for the background?

Reply: *The reason for the negligible background is unspecific disassembly of purified 70S during SDG, seen also in Figure 2.*

On the other hand they state that "Wildtype ABCE1 splits ribosomes most efficiently with ATP or AMP-PNP in an aRF1 dependent manner."

However, in the presence of ATP splitting is estimated to be less than 40% compared to the AMPPNP case. This is confusing.

Reply: *We have changed this sentence accordingly.*

3. The 50S/70S ratio as a measure for splitting (Fig. 3B) is given only normalized to the splitting with AMP-PNP, aRF1 and ABCE1. What is the absolute value? From Fig. 3 it seems that even in this 100% case the majority of the 70S ribosomes is not split. What is the reason? This requires explanation and discussion.

Reply: *Figure 3 extended view 3 clearly illustrates that 90% of our purified 70S can be split by ABCE1E238A or ABCE1E485A (as well as ABCE1E238A/E485A, which is not shown). We also mentioned on page 7 that substitution of a catalytic glutamate promotes ribosome splitting. The inability of ABCE1wt results from the experimental conditions (low temperature, high magnesium), which on the one hand must allow specific and on the other hand prevent unspecific dissociation. Our experimental conditions are common to archaeal translation assays and we provide references on page 7. The amount of 70S split by ABCE1wt corresponds to previous publications, a reference is provided on page 7. We further provide exemplary ratios for the SDG profiles presented in Figure 3 EV3 to facilitate understanding of our quantification method.*

4. As written, the asymmetry of the two nucleotide binding sites and the hyperactivity of ABCE1(E485A) caused by ATP occlusion in site II is presented as a novel finding. The finding of two asymmetric ATP-binding sites is even put into the Highlights. However, 7 years ago the Tampe group has already suggested functional and structural asymmetry of the two ATP-binding based on ATPase hyperactivity caused by the related E485Q mutant (Barthelme et al., 2011, PNAS). These related findings are not properly discussed. Barthelme et al., 2011 is cited several times throughout the manuscript for more general statements, but in the section "Two nucleotide-binding sites operate asymmetrically in ABCE1" no reference to this related paper can be found. This is not right.

Reply: *We have included the reference Barthelme et al, 2011, PNAS at the position suggested by the reviewer. However, we would like to point out that we investigate different mutations as in the reference, though with a similar result. As already mentioned, the strength of our manuscript is the detailed functional analysis of each mutant along the distinct steps of the ribosome recycling reaction, which deepens our understanding of this essential process.*

5. In Fig. 1C the k_{cat} for ATPase by wt ABCE1 is given as $\sim 20/\text{min}$. In Fig.3 extended view 1C a k_{cat} of $0.32/\text{min}$ is given for ABCE1 alone. What is the reason for this discrepancy?

Reply: As described in Material & Methods, stimulation of ATP hydrolysis by 70S and aRF1 was measured at splitting conditions, $45\text{ }^{\circ}\text{C}$ and in the presence of 25 mM Mg^{2+} ions, while ATPase activity of free ABCE1 is measured at physiological conditions for *S. solfataricus* ($70\text{ }^{\circ}\text{C}$, 2.5 mM Mg^{2+}). The high magnesium concentration and low temperature have inhibitory effects on ATPase activity of ABCE1 (as previously published). We have included this information in the text and respective figure legends.

6. The manuscript is very hard to read, especially for the non-expert reader. In general, it may help to elaborate on how certain conclusions have been derived. Especially for the figure legends, it would be helpful, if the figure content is better described.

Reply: We have revised our complete manuscript to facilitate readability.

The authors also should check the text for consistency. For example, on page 7 they first state that "ABCE1 splits ribosomes most efficiently with ATP or AMP-PNP" and then wonder that "it is surprising that even the substitution of both catalytic glutamates resulted in high splitting potential by the ATPase inactive ABCE1, thus demonstrating that ATP hydrolysis per se is not required to drive ribosome splitting". Why is this surprising? Isn't this expected from efficient splitting in the presence of the nonhydrolysable AMP-PNP?

Reply: Previous multiple-round recycling experiments in yeast and human systems demonstrated a strict dependency on ATP. We show that this dependency results from hydrolysis dependent ABCE1 release from the post-splitting complex, not from the molecular mechanism of ribosome splitting in particular and anticipate that this is indeed surprising for the ribosome field.

Reviewer #3:

In this study the interplay of the two individual ATP active sites of the ribosome recycling factor ABCE1 is detailed. By employing mutants that either block ATP-hydrolysis or ATP binding, the authors aim to detail the allosteric communication that takes place between the two domains. They begin by demonstrating the effect that blocking hydrolysis of either has on the turnover of the other. They see a dramatic enhancement of turnover when the catalytic residue of site II mutated. When paired with the observation that ATP binding mutants at site II lead to a clear decrease in ATP turnover, the authors conclude that ATP occupancy of this site serves a 'check-point' for the overall cycle.

They also show ATP turnover for double mutants of both the hydrolysis-and binding-residues. There is clearly more turnover in the hydrolysis double mutants, suggesting that residual activity still remains. No comment is made as to the limit of their detection. If this is genuine signal, than many of their further results are called into question, as ATP hydrolysis has not truly been abolished.

Reply: Taking into account the strong inhibition of ATPase activity by the double-substitution and the strong activation of 70S splitting and 30S binding activities, we consider our conclusions as fair, even though, as the reviewer has noticed correctly, residual ATPase activity can be detected for the double E-to-A variant. However, we have rephrased our analysis of the ATPase activity of this mutant.

The yeast results cannot be summarized due to figure quality.

Reply: We have adjusted the quality of the figure. As already mentioned in the reply to Reviewer #1, the diminished growth of yeast in our assay results from the specific experimental conditions, which are now included in the Material & Methods section.

Sucrose density gradients are utilized to show how nucleotide state influences formation of pre-splitting complexes. This passage is frustrating to follow, with many parenthetical notes and a seeming lack of consistency in nomenclature for both conditions assayed and how mutants are referenced.

Reply: We cannot follow the referee's arguments here as no examples or details are provided. The constant shuffling of the order that mutants in the figures are presented makes interpretation especially difficult, this is exacerbated with no consistent depiction of nucleotide condition.

Reply: We have changed the order of the mutants wherever it was appropriate. We agree that during superficial overview of the manuscript, the order of the S-to-R mutants might appear confusing since these mutations are found in the C-loop of the respective site on the opposing nucleotide binding domains. However, we included schematic thumbnails illustrating our ABCE1 variants. In the figures of the revised manuscript we added additional description of which site the mutants belong to.

This section manages to highlight the need to have ATP occupancy of site II in order to associate with the 70S ribosome.

The gradient profiles provided in this section are largely incomplete.

Reply: We can provide a profile for each blot. However, we believe that the figure would then be largely overloaded. The chosen profiles illustrate in Figure 2B the high splitting potential of the E-to-A mutants even in the presence of ADP and, in Figure 2C, the low splitting potential of the S-to-R mutants in the presence of AMP-PNP.

For example, the gradient for the double hydrolysis mutant in the presence of ADP is provided in 2B, yet the blot shown at the bottom is very poor quality.

Reply: The blot quality is satisfactory.

The blot in the presence of AMPPNP looks better, yet the gradient profile is not provided. No profiles are provided in the absence of nucleotide for any species as a control.

Reply: ADP is provided as negative control, as well as for ribosome splitting. ADP is an acknowledged negative control in the ABC field.

They go on to try and demonstrate the ability to split ribosomes is not a function of nucleotide hydrolysis, but of binding. Employing surplus IF6 prevents reassociation by binding the large subunit, allowing for the monitoring of a single round of splitting events. For WT ABCE1, splitting is optimal in the presence of AMPPNP and is diminished with ATP and ADP. It is confusing as to why there is such a drastic loss of splitting function for the two individual hydrolysis mutants in the presence of AMPPNP. One would expect that a non-hydrolysable nucleotide would elicit the same splitting function regardless of the ability of a subunit to carry out nucleotide hydrolysis. Does this suggest a contamination of ATP in the AMPPNP prep? Do the catalytic residues also influence affinity for nucleotide? They also state that "it is surprising that even the substitution of both catalytic glutamates resulted in high splitting potential by the ATPase inactive ABCE1." This cannot be said, as the supplement for figure 1 shows there is still hydrolysis.

Reply: We feel that the reviewer did not study the figure properly. The splitting efficiency of ABCE1E238A is significantly higher with AMP-PNP. The splitting activity of ABCE1E485A with AMP-PNP is diminished within the range of error.

The discussion around E485A and the double hydrolysis mutant is very confusing. They state that these species are "primed to adopt the fully closed conformation and induce ribosome splitting," with any nucleotide. Why? It is certainly clear that they are active under conditions where E238A is not, but without some more insightful piece of data this is something that needs to be omitted from the results.

Reply: A previous publication shows a preference for the closed state in the ABC-transporter MsbA solely due to the presence of analogous mutations of the catalytic base (Schultz et al., 2011). We have included this fact into the manuscript. We can fairly combine this previous finding with our results to draw a firm conclusion.

The nucleotide binding and recovery assays are presented in figure 4. This is a population level measurement however all data is presented as a single molecule result. It is not at all clear how the authors arrive at plot of figure 4A from their provided method. It is understandable that when blocking hydrolysis you will maximally occupy the two binding sites of the protein, but how is that quantification actually being done in the experiment. Is this just an assumption that is being made? It is again asserted that the double hydrolysis mutant is catalytically inactive despite already demonstrating that it does indeed have activity. This assay is done at very different conditions than their previous ATPase experiments. Has part of panel B been cut off?

Reply: This assay has been done at splitting conditions as explicitly stated in the results section on page 8, because the nucleotide occlusion state during the splitting itself is of particular interest for the manuscript. As mentioned before, the low temperature and high Mg²⁺ concentration have inhibitory effects on ATPase activity of ABCE1, thus, no activity is observed for

ABCE1E238A/E485A in this assay. The calculation and quantification procedure is described in detail in the Material & Methods section, which the reviewer probably did not read. However, we have included an additional Figure S2 to illustrate the quantification process.

The data provided in figure 5 suggests that nucleotide occupancy in site II is required for formation of the post splitting complex. This assay is performed in lysate, presumably there are intact 70s ribosomes present? Wouldn't including the entire density gradient be informative for the way the different nucleotide states for each mutant alter the total distribution of ABCE1 with different ribosome species

Reply: 70S from *S. solfataricus* are intrinsically unstable and not present in the lysate, a well-known fact about translation in crenarchaeota. A reference is given in the manuscript. We have additionally included the information on absent 70S into the figure legend.

1st Editorial Decision

28 May 2018

Thank you for submitting your revised manuscript entitled "Ribosome recycling is scheduled by the asymmetric action of two ATP-binding sites in ABCE1". Your manuscript was previously reviewed at a different journal, and you provided a revised version addressing the concerns obtained during this previous round of peer-review.

I appreciate the way you addressed the technical concerns raised by the reviewers, and I would be happy to publish your work in Life Science Alliance, pending some minor amendments:

- I understand that the yeast growth assay in Figure 1E was performed after only 14 hours of growth in order to show the dominant negative effects of site II catalytic base mutants of ABCE1 on yeast viability. I however agree with the reviewer that the figure is currently too difficult to interpret (especially for the (-) growth defect for the E247Q mutant), and I would like to therefore ask you to improve this figure. Maybe the original figure is of higher resolution and easier to interpret.

2nd Revision – Authors' response

31 May 2018

Responses to the Editors:

I understand that the yeast growth assay in Figure 1E was performed after only 14 hours of growth in order to show the dominant negative effects of site II catalytic base mutants of ABCE1 on yeast viability. I however agree with the reviewer that the figure is currently too difficult to interpret (especially for the (-) growth defect for the E247Q mutant), and I would like to therefore ask you to improve this figure. Maybe the original figure is of higher resolution and easier to interpret.

Reply: The Main Figure has been shortened and improved. An additional Appendix Figure S2 illustrates the method and the dominant negative effects.

2nd Editorial Decision

1 June 2018

Thank you for submitting your Research Article entitled "Ribosome recycling is coordinated by processive events in two asymmetric ATP sites of ABCE1". It is a pleasure to let you know that your manuscript is now accepted for publication in Life Science Alliance.

Congratulations on this interesting work.